# Impacts of Climate Change and Agricultural Practices on Nitrogen Processes, Genes, and Soil Nitrous Oxide Emissions: A Quantitative Review of Meta-Analyses

**Dafeng Hui \***[ID]**, Avedananda Ray, Lovish Kasrija and Jaekedah Christian**

Department of Biological Sciences, Tennessee State University, Nashville, TN 37209, USA; aray28@tnstate.edu (A.R.); lkasrija@tnstate.edu (L.K.); jchris10@tnstate.edu (J.C.)
* Correspondence: dhui@tnstate.edu; Tel.: +1-(615)-963-5777

**Abstract:** Microbial-driven processes, including nitrification and denitrification closely related to soil nitrous oxide ($N_2O$) production, are orchestrated by a network of enzymes and genes such as *amoA* genes from ammonia-oxidizing bacteria (*AOB*) and archaea (*AOA*), *narG* (nitrate reductase), *nirS* and *nirK* (nitrite reductase), and *nosZ* ($N_2O$ reductase). However, how climatic factors and agricultural practices could influence these genes and processes and, consequently, soil $N_2O$ emissions remain unclear. In this comprehensive review, we quantitatively assessed the effects of these factors on nitrogen processes and soil $N_2O$ emissions using mega-analysis (i.e., meta-meta-analysis). The results showed that global warming increased soil nitrification and denitrification rates, leading to an overall increase in soil $N_2O$ emissions by 159.7%. Elevated $CO_2$ stimulated both *nirK* and *nirS* with a substantial increase in soil $N_2O$ emission by 40.6%. Nitrogen fertilization amplified $NH_4^+$-N and $NO_3^-$-N contents, promoting *AOB*, *nirS*, and *nirK*, and caused a 153.2% increase in soil $N_2O$ emission. The application of biochar enhanced *AOA*, *nirS*, and *nosZ*, ultimately reducing soil $N_2O$ emission by 15.8%. Exposure to microplastics mostly stimulated the denitrification process and increased soil $N_2O$ emissions by 140.4%. These findings provide valuable insights into the mechanistic underpinnings of nitrogen processes and the microbial regulation of soil $N_2O$ emissions.

**Keywords:** denitrification; global warming; greenhouse gas emission; mega-analysis; nitrogen fertilizer; $N_2O$; precipitation

## 1. Introduction

In the face of a growing global population, a paramount challenge is to increase production levels of food, feed, fiber, and fuel crops while simultaneously mitigating associated environmental impacts [1–3]. To meet the ever-increasing demands for food and energy, substantial quantities of chemical fertilizers, notably inorganic nitrogen (N) fertilizers, are routinely applied to agricultural lands each year. Although essential for production, this practice has created a serious problem: the release of soil greenhouse gases, most notably nitrous oxide ($N_2O$), into the atmosphere [4,5]. The repeated and excessive use of N fertilizers, coupled with N deposition and climate change, has amplified challenges related to nitrate leaching and $N_2O$ emissions. Agricultural soils contribute up to 80% of anthropogenic $N_2O$ emissions [6–8]. Remarkably, $N_2O$ is a potent, long-lived powerful greenhouse gas with a global warming potential 265 times greater than $CO_2$ [2,9]. In addition to its impact on global warming, $N_2O$ plays a significant role in stratospheric $O_3$ depletion [10]. Through photolysis and oxidation to nitric oxide, $N_2O$ can contribute to $O_3$ depletion in the stratosphere, further accelerating global climate change with diverse effects on human health [11,12].

In recent years, the atmospheric $N_2O$ concentration has risen from 270 ppb during the preindustrial era to 330 ppb, with an average increase of 0.73 ppb year$^{-1}$ [2,13]. Global $N_2O$ emissions stemming from N inputs have surged by more than 30% in the past four

decades [7,14]. Projections suggest that by 2030, $N_2O$ emissions from croplands could make up 59% of global $N_2O$ emissions [5,15]. This heightened $N_2O$ emission disrupts greenhouse gas balances, offsetting the climate benefits gained from $CO_2$ removal and other climate mitigation strategies [2,16]. To address this escalating $N_2O$ issue, a comprehensive understanding of the mechanisms and mitigation strategies for soil $N_2O$ emission is not just valuable but indeed imperative.

Soil $N_2O$ production and soil N cycling are intricately influenced by a diverse array of functional soil microorganisms [5,17,18]. Key players in this context include *amoA* genes of ammonia-oxidizing archaea (*AOA*) and ammonia-oxidizing bacteria (*AOB*), along with crucial functional genes such as *narG* (encoding nitrate reductase), *nirK* and *nirS* (encoding nitrite reductase), and *norB* and *nosZ* (encoding nitrous oxide reductase) genes [5,8,19]. These genes are significant actors in soil nitrification and denitrification processes. It is important to note that these nitrifying and denitrifying genes can be influenced by many factors, including climate change and various agricultural practices, leading to modifications in soil N transformation rates [5,20,21]. Consequently, evaluating the impacts of global climate change and agricultural practices on N cycling, especially concerning nitrification and denitrification, holds significant importance, as their effects on these microbial processes can induce positive feedback on climate change [7,22].

Numerous investigations have been undertaken in recent decades to explore the repercussions of climate change and agricultural practices on soil $N_2O$ emissions in terrestrial ecosystems [2,23,24]. Due to the inconsistency among different individual studies, meta-analysis has been utilized to synthesize the results from these studies. Recently, a surge in meta-analyses has sought to quantify the impacts of these factors and practices [5,8,14,22]. However, these meta-analyses have, at times, produced varying results, promoting the need for a comprehensive evaluation [25]. This review aims to fill this gap by synthesizing the results of these meta-analyses on the impacts of climate change and agricultural practices on soil $N_2O$ emissions. It also delves into potential mechanisms underlying these effects on enzyme activities and genes associated with nitrification and denitrification processes. This review begins with a brief overview of N cycling and the role of $N_2O$ emissions in climate change, laying the groundwork for an in-depth discussion on the process and key genes governing soil emissions. Special emphasis is placed on recent meta-analysis studies, quantifying the impacts of climate change and agricultural practices, such as global warming, elevated $CO_2$, precipitation changes, N fertilization, and biochar application on soil $N_2O$ emissions, utilizing mega-analysis (i.e., meta-meta-analysis) techniques. The primary goal of this review is to shed light on existing knowledge while identifying domains deserving of further investigation, emphasizing the significance of ongoing research in this crucial field.

## 2. Nitrogen Processes, Enzymes, Genes, and Soil $N_2O$ Emissions

Soil N cycling and transforming processes in terrestrial ecosystems are predominantly regulated by soil microorganisms, with their functional genes and their extracellular enzymes playing a central role in these processes [26–29]. Soil $N_2O$ is produced from microbial activities, involving archaea, bacteria, and fungi, engaged in the conversion of inorganic N through a series of processes. Typically, these processes encompass nitrification, which is the aerobic oxidation of $NH_4^+$ to $NO_3^-$ via $NO_2^-$, and denitrification, the anaerobic reduction of $NO_3^-$ to $N_2O$ and $N_2$. This intricate cycle is tightly regulated by a spectrum of enzymes and multiple functional genes [8,30] (Figure 1).

As soil N progresses through the biogeochemical cycle in terrestrial ecosystems, it starts with crucial processes: N fixation and mineralization [31]. These processes are driven by the associated microbial communities [31]. During soil biological N fixation, molecular N is reduced to ammonia through specific biological enzymes [31,32]. Non-symbiotic N fixation, driven by free-living diazotrophs, emerges as a primary source of N in terrestrial ecosystems [33]. The key marker gene in this process is *nifH*, which encodes the nitrogenase reductase subunit responsible for reducing nitrogen gas ($N_2$) to ammonium ($NH_4^+$) [34,35].

While most N-fixation occurs within the root nodules of legumes via symbiotic bacteria, free-living N fixation serves as a potential source for biological N inputs in non-leguminous crops [27,33]. Regarding mineralization, the N-cycling enzymes in soil microbes regulate inorganic N availability via mineralization and hydrolysis [29,36]. Key enzymes (and marker genes) involved in N mineralization include protease (*npr* and *sub*), chitinase (*chiA*), urease (*ureC*), and arginase (*rocF*) [37].

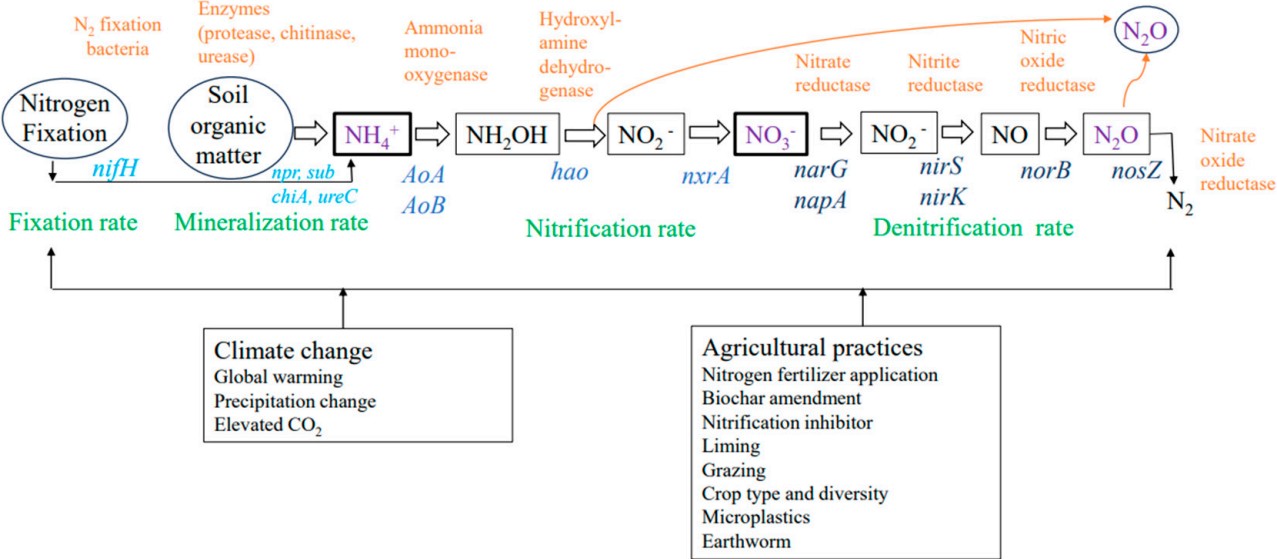

**Figure 1.** Nitrogen processes, genes, enzymes, soil N₂O emissions, and impact factors investigated in this study [13].

Nitrification is a fundamental process in which inorganic reduced N undergoes oxidation to become nitrate N under the action of ammonia-oxidizing microorganisms (Figure 1) [31,38]. Key genes involved in this process include *amoA*, which codes for a subunit of ammonia monooxygenase. This enzyme is pivotal for the transformation of ammonia ($NH_3$) or $NH_4^+$ to hydroxylamine [31,35]. In particular, this first and rate-limiting step of nitrification, namely ammonia oxidation, is facilitated by either *AOA* (ammonia-oxidizing archaea) or *AOB* (ammonia-oxidizing bacteria). Both groups carry the *amoA* gene [8,12,39,40].

Denitrification is a critical process involving the reduction of nitrate ($NO_3^-$) to nitrite ($NO_2^-$) (Figure 1). Under anaerobic conditions, denitrifying bacteria reduce $NO_3^-$ or $NO_2^-$ to gaseous N, such as nitric oxide (NO) and N₂O [31,41]. This pathway is a major source of soil N₂O production. Following the reduction of $NO_3^-$ to $NO_2^-$, two potential pathways can be taken. It can be transformed into $NH_4^+$ through dissimilatory reduction or further reduced to N₂O [35,42]. Various denitrification genes play important roles in regulating N₂O production. The denitrification process involves genes such as *narG* and *napA*, which encode subunits of two distinct nitrate reductases: membrane-bound nitrate reductase (NAR) and periplasmic nitrate reductase (NAP) [40]. *napA* is also considered a part of the assimilatory N reduction pathway. These enzymes mediate the reduction of $NO_3^-$ to $NO_2^-$ in both denitrification and dissimilatory $NO_3^-$ reduction to $NH_4^+$ [40]. Subsequently, the *nirK* and *nirS* genes, encoding nitrite reductase, are indicators of denitrifiers that facilitate the reduction of $NO_2^-$ to NO, which is a rate-limiting step in denitrification [9,43,44]. The *norB* gene, encoding for the NO reductase, facilitates the reduction of NO to N₂O. Finally, the *nosZ* gene, encoding N₂O reductase, catalyzes the transformation of N₂O into N₂, representing the final step in denitrification [12,45]. These genetic markers provide insights into the complex processes of denitrification [9,31,40]. Understanding alterations in the abundance and diversity of these N functional genes provides insights into the biotic mechanisms mediating N₂O emissions [9].

Quantifying and characterizing these functional genes associated with the N biogeochemical cycle establishes a direct link between N-cycling microbial groups and the actual N processes, enriching our understanding of the ecological importance of N-cycling traits and soil $N_2O$ emissions [27,35,40,46]. For example, nitrification inhibitors can limit the growth and activity of nitrifiers by deactivating the ammonia monooxygenase enzyme [30,47]. Elevated temperatures enhance nitrification and denitrification rates, thereby impacting soil N pools and N availability to plants [28]. These processes are intricately regulated through the interplay of various factors, including temperature, precipitation or soil moisture content, N availability, N fertilization rates, soil microbial biomass, pH, and oxygen supply [7,48] (Figure 1). The complexity of the mechanisms controlling $N_2O$ emissions makes it challenging to predict how climate change and agricultural practices will influence these emissions. To gain deeper insights into the mechanisms responsible for soil $N_2O$ emission under changing climate and evolving agricultural practices, it is essential to enhance our understanding of how soil microbial communities engaged in soil N cycling respond to these dynamic factors [9].

## 3. Methods of Study: Experimental Study, Meta-Analysis, and Mega-Analysis

To address the intricate impacts of climate change and agricultural practices on N processes, genes, and the regulation of soil $N_2O$ emissions, numerous field and laboratory studies have been carried out [25,40,49]. For example, Xu et al. [50] reported that warmer and drier conditions have led to reduced $N_2O$ emissions, with soil $N_2O$ being closely associated with the abundance of *AOB*, *nirK*, and *nosZ* genes. Meanwhile, Huang et al. [51] demonstrated a 41.83% reduction of soil $N_2O$ emission induced by a biological nitrification inhibitor, attributing the reduction to the promotion of bacteria with the *nosZ* gene, while the growth of bacteria with *nirS* and *nirK* genes was inhibited. While these studies have contributed valuable insights, the presence of disparities and contradictions among individual findings underscores the need for a systematic approach.

Meta-analysis, a statistical method that combines the results of multiple individual studies to generate an overall effect size, proves indispensable in resolving these discrepancies in ecological studies [52,53]. This approach provides a more robust estimation and helps reconcile conflicting results from individual studies. Meta-analysis has been widely applied in ecological studies to quantify the effects of climate change and agricultural practices on soil $N_2O$ emissions [30,40]. However, the proliferation of meta-analyses has led to inconsistent findings and occasionally opposing conclusions, necessitating a critical evaluation and synthesis of existing meta-analyses [25,54–56].

Recently, Kaur et al. [25] conducted a critical assessment of 18 meta-analyses concerning the impacts of biochar application on soil $N_2O$ emission. Their findings revealed that the reduction of soil $N_2O$ emission ranges from no change to a substantial reduction of 58%. They further applied the approach of mega-analysis, estimating an overall reduction of 38% by synthesizing data from these 18 meta-analyses. Mega-analysis, or a meta-analysis of meta-analyses, offers a more precise estimate of the mean effect size by combining data from various meta-analyses [25,54]. This approach has been found to provide effect size quantification with low bias and high precision [55].

In this study, we conducted a comprehensive literature search for meta-analysis studies on the impacts of climate change and agricultural practices on soil N process, genes, and $N_2O$ emission. Utilizing the keywords "meta-analysis" and "$N_2O$ emission", we identified 326 relevant papers through the Web of Science (Figure S1). We carefully filtered these papers, excluding those that were not true meta-analyses, such as individual field experiments, modeling studies, studies that investigated factors unrelated to climate change or agricultural practices, and those that did not cover the N processes, enzymes, and genes involved in N fixation, mineralization, nitrification, or denitrification. In total, we included 25 meta-analysis studies in this review. For studies addressing multiple meta-analyses on the impacts of climate change factors and agricultural practices, a mega-analysis was conducted to consolidate findings. Following Kaur et al. [25], the grand mean

of response ratio was calculated using the weighted mean of response ratio using the following equation:

$$RR = \frac{\sum(ni * RRi)}{\sum ni} \tag{1}$$

where *RR* is the grand response ratio of response variables such as genes and soil $N_2O$ emissions across all studies, *RRi* is the response ratio of these variables from the *i*th meta-analysis study, and $n_i$ is the sample size. The 95% confidence interval of the grand mean of *RR* was calculated using the weighted standard error. The *RR*, standard error, and 95% CI were further converted to percentage change using the following equation:

$$RR\ (\%)\ =\ (e^{lnRR}-1) \times 100 \tag{2}$$

## 4. Impacts of Climate Change on Gene, Enzyme, Nitrification and Denitrification Processes, and Soil $N_2O$ Emissions

Climate change such as global warming, elevated $CO_2$ concentration, and precipitation change has the potential to exert significant influences on soil N processes and soil $N_2O$ emissions [57–59]. Here, we collected data from meta-analyses exploring the impacts of climate change and quantitatively evaluated the response ratios concerning the effects of global warming, elevated $CO_2$ levels, alterations in precipitation patterns on various aspects of soil N dynamics, including soil N pools, abundance of genes, and soil $N_2O$ emissions.

### 4.1. Impacts of Global Warming

Global warming has the potential to significantly influence microbially mediated N cycling processes, including N mineralization, nitrification, and denitrification. These alternations can lead to notable changes in soil N pool sizes, N availability, and soil $N_2O$ emissions in terrestrial ecosystems [28]. A range of studies has demonstrated that increases in temperature can modify microbial N immobilization and mineralization rates [60–62]. Ecosystems and climate zones may have significant impacts on soil N processes. For example, those in forest soils can have different responses from those in grasslands. Cold regions may show more sensitive responses to warming than warm regions [48]. Additionally, several studies have reported that increased temperatures stimulate soil microbial metabolism, enhance soil enzyme activities, and accelerate the decomposition of organic matter [59]. Furthermore, elevated temperatures have been found to increase the abundances of genes like *nirK* and *nosZ* [63,64], with *nirS*-containing denitrifiers being more sensitive to temperature increases than those containing *nirK* and *nosZ* genes [57]. However, it is worth noting that several other studies have reported that elevated temperatures do not change the abundance of *amoA* genes or have found inconsistent responses of *AOA* and *AOB* to elevated temperatures [65,66].

To assess the overall impact of global warming on N processes, meta-analyses have been conducted on the impacts of temperature on soil N processes, enzyme activities, and soil functional genes involved in $N_2O$ emission. In an early meta-analysis examining the influence of warming on soil $N_2O$ emission, Bai et al. [67] collected 528 observations from 51 papers, revealing a non-significant mean effect size of 0.128 of soil $N_2O$ emission by warming, based on 26 studies. Dai et al. [28] synthesized a comprehensive dataset of 1270 observations from 134 papers and revealed that elevated temperature significantly amplifies soil nitrification and denitrification rates, leading to a notable surge of up to 227% in $N_2O$ emissions. The prevalence of the *nirS* gene increases in the presence of plants, whereas the *nosZ* gene becomes more predominant in the absence of plants at elevated temperatures. Conversely, the *AOA*, *AOB*, and *nirK* genes remain unaffected by the elevated temperature. More recently, Li et al. [58] analyzed 72 case studies from 46 papers and found that increased temperatures do not significantly affect the abundance of archaeal *amoA*, bacterial *amoA*, and *nosZ* genes, but they significantly decrease the abundances of *nirK* and *nirS* genes by 26% and 31%, respectively. Temperature increases $N_2O$ emissions by 33%. Additionally, Salazar et al. [49] found that warming leads to increased N mineralization rates and $N_2O$

emissions in cold ecosystems due to heightened enzyme activity targeting relatively labile N sources rather than alternations in the abundance of N-relevant genes (e.g., *amoA* and *nosZ*). Liu et al. [68] synthesized 1845 measurements from 164 publications and found that warming significantly enhances soil $N_2O$ emission. About 1.5 °C of experimental warming significantly stimulates $N_2O$ emissions by 35.2%.

We quantified the impacts of warming on nitrification and denitrification genes and soil $N_2O$ emission based on four meta-analyses (Tables 1 and 2). The results of this mega-analysis showed that warming did not influence *AOA* or *AOB* (Table 2), but reduced MBN (−15.1%), and stimulated soil $N_2O$ emissions (147.9%) (Table 1). Warming did not change *nirK*, *nirS*, and *nosZ* (Table 2). It enhanced the mineralization rate by 153.0%, nitrification rate by 62.0%, and denitrification rate by 159.7% (Table 1). It also increased Protease by 38.7% and Urease by 216.5% (Table 1). While there was no significant impact on the abundance of *AOA* and *AOB*, warming led to a decrease in MBN, increased soil $N_2O$ emissions, and stimulated rates of N cycling processes such as mineralization, nitrification, and denitrification. The increased enzyme activities further highlight the accelerated decomposition of organic matter and nutrient cycling under warming conditions.

**Table 1.** Effects of climate change factors and agricultural practices on the main microbial enzymes and nitrification and denitrification rates, based on results of meta- and mega-analyses. Values are effective size (RR, %) with 95% confidence intervals.

| Treatment | Nitrogen Fixation | | Mineralization | | | | Nitrification | | | Denitrification | | MBC | MBN | Soil $N_2O$ Emission | References |
|---|---|---|---|---|---|---|---|---|---|---|---|---|---|---|---|
| | Rate | Bacteria | Rate | Protease | Urease | $NH_4^+$-N | Rate | Enzyme | $NO_3^-$-N | Rate | Enzyme | | | | |
| **Climate change** | | | | | | | | | | | | | | | |
| Warming | | −9.1% [−34.7%, 26.5%] | **153.0% [106.9%, 209.4%]** | **38.7% [6.4%, 80.7%]** | **216.5% [59.5%, 528.0%]** | −0.6% [−20.8%, 24.7%] | **62.0% [33.2%, 97.1%]** | | 8.6% [−14.0%, 37.4%] | **159.7% [127.1%, 196.9%]** | | | **−15.1% [−27.4%, −0.7%]** | **147.9% [92.2%, 219.7%]** | [28,49,58,59] |
| Elevated $CO_2$ | | | | | | −0.3% [−7.1%, 7.0%] | **32.7% [7.4%, 63.9%]** | **−18.4% [−32.7%, −1.0%]** | **13.1% [4.7%, 22.1%]** | 5.3% [−7.7%, 20.1%] | **−20.8% [−34.5%, −4.2%]** | 1.6% [−20.1%, 29.1%] | **27.8% [16.4%, 40.3%]** | **40.6% [25.3%, 57.8%]** | [12,22,59] |
| PPT+ | | | | | | | | | | | | | | **54.2% [29.5%, 83.7%]** | [58,59] |
| PPT− | | | | | | | | | | | | | | **−45.9% [−55.9%, −33.6%]** | [58,59] |
| **Agricultural practices** | | | | | | | | | | | | | | | |
| Nitrogen fertilization | | | −3.0% [−11.3%, 6.2%] | **27.1% [15.0%, 40.5%]** | **81.1% [67.3%, 96.2%]** | **33.9% [17.4%, 52.9%]** | **198.0% [98.4%, 347.8%] 216.0% [99.5%, 400.5%]** | | **94.2% [68.4%, 123.9%]** | **42.0% [9.6%, 84.0%]** | **28.5% [2.4%, 61.5%]** | 113.7% [−87.2%, 3463%] | **28.4% [7.4%, 53.6%]** | **153.2% [109.6%, 205.9%]** | [9,27,35,69–72] |
| Biochar addition | **13.2% [2.9%, 24.6%]** | **44.4% [29.4%, 61.1%]** | **14.4% [1.2%, 29.3%]** | | 4.0% [−18.2, 32.3%] | **6.0% [1.0%, 11.3%]** | **40.8% [1.8%, 94.8%]** | | **3.4% [0.4%, 6.5%]** | 13.3% [−2.9%, 32.2%] | 27.3% [−0.4%, 62.7%] | | **13.2% [8.3%, 18.3%]** | **−15.8% [−20.5%, −10.8%]** | [14,40,72] |
| Nitrification inhibitor | | | | | | **87.8% [68.2%, 109.6%]** | **−21.3% [−26.6%, −15.7%]** | **−32.8% [−38.8%, −26.4%]** | **−37.7% [−41.9%, −33.3%]** | | **−25.3% [−34.1, −15.2%]** | | | **−56.1% [−64.8%, −45.0%]** | [5,30,72] |
| Liming | | −20.8% [−39.4%, 3.6%] | | | | −3.1% [−24.3%, 23.8%] | **62.7% [27.6%, 107.4%]** | | **55.8% [35.2%, 79.6%]** | **201.1% [94.7%, 365.7%]** | | | | **−37.9% [−48.2%, −25.5%]** | [73] |
| Microplastics | | | 4.9% [−9.9%, 22.1] | **149.2% [112.0%, 192.8%]** | **6.8% [2.1%, 11.7%]** | **−6.8% [−12.8%, −0.4%]** | 0.9% [−2.5%, 4.5%] | | **−22.3% [−28.3%, −16.0%]** | **17.8% [1.4%, 36.8%]** | | | **27.5% [4.4%, 55.8%]** | **140.4% [73.7%, 232.7%]** | [74] |

**Table 1.** *Cont.*

| Treatment | Nitrogen Fixation | | Mineralization | | | | | Nitrification | | | Denitrification | | MBC | MBN | Soil N₂O Emission | References |
|---|---|---|---|---|---|---|---|---|---|---|---|---|---|---|---|---|
| | Rate | Bacteria | Rate | Protease | Urease | NH₄⁺-N | Rate | Enzyme | NO₃⁻-N | Rate | Enzyme | | | | |
| Crop diversity | | | −6.6% [−21.1%, 10.5%] | | | −11.4% [−23.4%, 2.4%] | **−24.4% [−35.1%, −12.0%]** | | −6.8% [−24.6%, 15.1%] | −24.6% [−49.1%, 11.7%] | | | | −19.9% [−32.0%, −5.6%] | [31] |
| Grazing | | | −6.6% [−21.1%, 10.5%] | | | −11.4% [−23.4%, 2.4%] | **−24.4% [−35.1%, −12.0%]** | | −6.9% [−24.6%, 15.1%] | −24.6% [−49.1%, 11.7%] | | | | **−19.9% [−32.0%, −5.6%]** | [8] |
| Earthworm | | | **217.8% [45.5%, 594.1%]** | | | **71.2% [31.6%, 122.6%]** | 58.5% [−40.9%, 324.7%] | | 17.9% [0.4%, 38.6%] | **38.6% [17.8%, 63.0%]** 0.3 [0.2,0.5] | | | **13.7% [0.3%, 28.8%]** | **28.9% [0, 68.0%]** | [75] |

Note: Bolded values indicate a significant effect. PPT+ indicates increased precipitation. PPT− indicates reduced precipitation. MBC: microbial biomass carbon. MBN: microbial biomass nitrogen.

**Table 2.** Effects of climate change factors and agricultural practices on the abundance of main genes involved in the processes of nitrification and denitrification based on the results of meta- and mega-analyses. Values are effective size (RR, %) with 95% confidence intervals.

| Treatment | Nitrogen Fixation | Nitrification | | Denitrification | | | | | Reference |
|---|---|---|---|---|---|---|---|---|---|
| | *nifH* | *AOA* | *AOB* | *narG* | *nirS* | *nirK* | *norB* | *norZ* | |
| **Climate change** | | | | | | | | | |
| Warming | −5.1% [−11.3%, 6.7%] | 0.7% [−15.5%, 20.0%] | −1.34% [−22.0%, 24.7%] | | 12.7% [−2.3%, 30.0%] | −9.7% [−23.5%, 6.7%] | | 50.9% [−6.9%, 144.6%] | [28,49,58,59] |
| Elevated CO$_2$ | | 5.7% [−9.8%, 23.9%] | 12.8% [−9.5%, 40.6%] | | **18.0% [0.4%, 38.6%]** | 18.6% [−2.8%, 43.3%] | | 2.6% [−16.9%, 26.7%] | [12,22,59] |
| PPT+ | | −5.1% [−29.9%, 28.4%] | −33.2% [−44.7%, 9.4%] | | 31.2% [−9.0%, 89.1%] | −5.0% [−20.9%, 14.2%] | | 4.2% [−15.6%, 28.5%] | [58,59] |
| PPT− | | 28.5% [−20.0%, 106.6%] | 23.4% [−4.1%, 58.8%] | | **173.6% [39.5%, 436.5%]** | 79.6% [−8.0%, 250.6%] | | 48.7% [−19.6%, 175.0%] | [58,59] |
| **Agricultural practices** | | | | | | | | | |
| Nitrogen fertilization | −5.9% [−23.5%, 15.6%] | **15.0% [−0.8%, 33.3%]** | **146.3% [108.9%, 190.4%]** | | **31.8% [6.5%, 63.0%]** | **43.2% [12.2%, 82.8%]** | | 38.2% [−6.9%, 105.1%] | [9,27,35,69–72] |
| Biochar addition | 4.7% [−9.8%, 21.6%] | **23.0% [10.1%, 37.4%]** | 11.0% [−4.3%, 28.7%] | **17.2% [0.2%, 37.1%]** | **19.1% [6.2%, 33.7%]** | **28.3% [13.1%, 45.4%]** | | **17.1% [7.5%, 27.6%]** | [14,40,72] |
| Nitrification inhibitor | | −6.3% [−14.0%, 2.1%] | **−51.9% [−61.5%, −40.0%]** | −4.3% [−52.5%, 93.1%] | **−22.7% [−37.9%, −3.9%]** | **−20.0% [−28.7%, −10.3%]** | | −0.04% [−16.3%, 19.7%] | [5,30,72] |
| Liming | | **70.2% [16.2%, 149.2%]** | **132.6% [35.2%, 299.9%]** | | **37.5% [9.9%, 72.0%]** | **142.0% [54.3%, 279.6%]** | | 16.0% [−1.5%, 36.5%] | [73] |
| Microplastics | | 0.4% [−11.2%, 14.2%] | | **−5.2% [−8.7%, −1.5%]** | **18.6% [5.1%, 34.0%]** | | **−36.8% [−55.5%, −10.1%]** | **10.6% [2.7%, 19.1%]** | [74] |
| Crop diversity | **33.8% [17.3%, 52.2%]** | 18.2% [−1.4%, 41.6%] | −2.8% [−12.7%, 8.3%] | **17.8% [5.1%, 32.2%]** | **39.4% [18.5%, 63.9%]** | **14.0% [0.5%, 29.3%]** | | 3.8% [−8.4%, 17.5%] | [31] |
| Grazing | | **−34.9% [−57.5%, −1.99%]** | −28.5% [−57.3, 19.5%] | **−28.3% [−40.5%, −14.0%]** | **−35.3% [−55.3%, −6.1%]** | −3.4% [−61.8%, 142.7%] | | −22.1% [−42.2%, 4.9%] | [8] |
| Earthworm | 27.3% [−7.2%, 74.8%] | 269.6% [−34.1%, 1974.4%] | **10.8% [1.3%, 21.1%]** | | | −12.3% [−32.6%, 14.2%] | | 5.1% [−8.0%, 21.0%] | [75] |

Note: Bolded values indicate significant effects. PPT+ indicates increased precipitation. PPT− indicates reduced precipitation. *AOA*: *amoA* genes from ammonia-oxidizing archaea. *AOB*: *amoA* genes from ammonia-oxidizing bacteria. *narG*: nitrate reductase. *nirS* and *nirK*: nitrite reductase. *nosZ*: N$_2$O reductase.

*4.2. Impacts of Elevated $CO_2$*

It is well documented that elevated $CO_2$ enhances plant growth and biomass production and increases ecosystem carbon sequestration [13,53,58]. Elevated $CO_2$ can also have several other impacts, such as promoting organic C decomposition, enhancing microbial activity, and stimulating soil extracellular enzyme activity [59,76]. However, the impacts of elevated $CO_2$ on nitrification, denitrification, and associated functional genes are still a topic of ongoing research, and the results are not consistent. Various studies have reported diverse responses to elevated $CO_2$ in the nitrification and denitrification rates, with some showing negative, positive, or neutral effects of elevated $CO_2$ on these processes [13,77]. Variations also exist in the responses of nitrifying and denitrifying functional genes to elevated $CO_2$ conditions. Different studies have reported divergent outcomes, indicating that the amounts of *AOA*, *AOB*, *nirK*, *nirS*, and *nosZ* functional genes may exhibit increases, decreases, or remain unaffected under elevated $CO_2$ levels [77–79].

We found that four meta-analyses have been published on the impacts of elevated $CO_2$ on genes involved in N processes and $N_2O$ emissions. Barnard et al. [80] reviewed the impacts of elevated $CO_2$, N, and temperature on nitrification, denitrification, and soil $N_2O$ emission and found that elevated $CO_2$ enhanced net nitrification, reduced potential denitrification ($-18\%$), increased net nitrification (33%), and did not significantly alter soil $N_2O$ emissions. Li et al. [59] analyzed the impacts of multiple climate factors on N-cycling genes and found that elevated $CO_2$ increased N-cycling functional gene abundances (19.5%). In particular, elevated $CO_2$ increased *nirK* but did not change *AOB*. Du et al. [12] collected data from 50 publications and reported that elevated $CO_2$ enhanced $N_2O$ emissions by 44%. Elevated $CO_2$ increases the abundance of *AOB* (21%), *nirK* (15%), and *nirS* (15%) but does not change *AOA* and *nosZ*. Gineyts and Niboyet [22] used 879 observations from 58 papers and found that elevated $CO_2$ increased *AOA* (62%), *nirK* (32%), and *nirS* (27%), leading to 26% increases in soil $N_2O$ emission.

Synthesizing these meta-analyses, our results showed that elevated $CO_2$ increased the abundance of *nirS* (18.0%) and soil $N_2O$ emission by 40.6% but did not significantly change *AOA*, *AOB*, *nirK*, and *nosZ* (Tables 1 and 2). Elevated $CO_2$ also increased MBN (27.8%), net nitrification rate (32.7%), and $NO_3^-$-N (13.1%) but reduced nitrifying enzymes by 18.4% [12,22,59,80] (Table 1). While there was an increase in soil $N_2O$ emissions, possibly associated with changes in denitrification (as indicated by increased *nirS* abundance), there were mixed effects on nitrification-related parameters. The increased MBN, net nitrification rate, and $NO_3^-$-N levels indicate the stimulation of N cycling processes, but the reduction in nitrifying enzymes suggests a potential deceleration of ammonia oxidation. These findings highlight the need to consider multiple factors influencing soil N dynamics in the context of elevated $CO_2$.

*4.3. Impacts of Precipitation*

In terrestrial ecosystems, precipitation change can have multifaceted effects. These alterations influence soil microclimate and impact the soil water balance, soil aeration, nutrient availability, and microbial ecology [58,67]. Consequently, they play a role in shaping soil $N_2O$ emissions. For example, Šťovíček et al. [81] found that soil microbial diversity tends to be high under dry conditions due to the fragmentation of niches in dry soils. However, drought can also reduce the genetic potential and stability of soil microbiomes [59]. Homyak et al. [82] investigated the effects of reduced precipitation on soil $N_2O$ emissions and found that a reduction in precipitation significantly lowers soil $N_2O$ emissions, suggesting that denitrification is more sensitive to drought than processes controlling N supply. Decreased precipitation appears to have minimal effects on the abundances of archaeal *amoA*, bacterial *amoA*, *nirK*, and *nosZ*, but it shows positive effects on the abundances of *nirS* [58]. Conversely, increased precipitation has little effect on the abundances of archaeal *amoA*, *nirK*, *nirS*, and *nosZ* while exhibiting negative effects on the abundances of bacterial *amoA* [58].

Several meta-analyses have been conducted to further understand the impacts of precipitation on soil $N_2O$ emissions [58,83]. For example, Yan et al. [83] performed a meta-analysis incorporating 84 published studies and found that increased precipitation significantly increases $N_2O$ emissions (+154.0%), whereas decreased precipitation significantly decreases $N_2O$ emissions ($-$64.7%). They also found that precipitation increases enhanced soil $N_2O$ emissions by 128.3% in temperate forests and by 179.6% in boreal forests but did not influence soil $N_2O$ emissions in grasslands. The impacts of decreased precipitation also varied in different ecosystems, ranging from no effect in subtropical forests to $-$24.3% in temperate forests and $-$92.6% in grasslands. However, only two papers have synthesized the impacts of precipitation on N-cycling genes and soil $N_2O$ emission. Li et al. [58] explored the effect of global climate change on $N_2O$ emissions and the related N functional genes in terrestrial ecosystems. Their findings indicated that precipitation promoted $N_2O$ emissions by 55%, while reduced precipitation inhibited $N_2O$ emissions by 31%. Based on two meta-analyses, our results showed that increased precipitation did not influence *AOA*, *AOB*, *nirkS*, *nirK*, or *nosZ* (Table 2) [58,59]. Reduced precipitation did not change *AOA* and *AOB* but increased *nirS* by 173.6% and reduced soil $N_2O$ emission by 45.9% (Tables 1 and 2) [58,59]. It is worth noting that the sample sizes of two meta-analyses were also small (18 and 20 for soil $N_2O$ emissions). More studies are needed to focus on the N-cycling genes. Nevertheless, the findings emphasize the importance of considering the direction and magnitude of precipitation changes when assessing its impact on soil N dynamics and greenhouse gas emissions.

## 5. Impacts of Agricultural Practices on Gene, Enzyme, Nitrification and Denitrification, and Soil $N_2O$ Emissions

Various agricultural management practices, including N fertilizer application, conservation tillage, cover cropping, soil amendments like biochar, and the utilization of nitrification inhibitors, can significantly affect soil physiochemical factors and microbial activity. These practices, in turn, have the potential to influence soil nitrification and denitrification processes, ultimately leading to changes in soil $N_2O$ emissions. Notably, the application of N fertilizers, the incorporation of biochar into the soil, and the use of nitrification inhibitors are of particular importance in this context. Here, we focused on understanding the effects of these and other agricultural practices on N processes, genes, and soil $N_2O$ emissions.

### 5.1. Impacts of Nitrogen Application

Nitrogen (N), an essential element, serves as a critical determinant of plant growth and productivity in terrestrial ecosystems [9,84,85]. The application of N fertilizer, a fundamental agricultural practice, has made a substantial impact on plant biomass production, soil microbial activities, and soil $N_2O$ emissions [35,86]. The response of N cycling gene abundances to N fertilization is influenced by several factors [27]. For example, different sources of N fertilization, such as mineral and organic N fertilization, often lead to distinct changes in N-cycling gene abundances [27,32,71,87]. Additionally, N deposition increases the availability of soil nutrients and can cause soil acidification. This, in turn, affects soil element stoichiometry, nutrient utilization, and limitation, thus influencing soil microbial communities [70,88]. N deposition also has effects on functional genes related to N cycling [70]. Different studies have drawn disparate conclusions regarding the impact of functional genes and external environmental factors on $N_2O$ emissions following N addition. For example, Soares et al. [89] found that $N_2O$ emissions are associated with the abundance of *AOA* but exhibit no correlation with the *nirK*, *nirS*, and *nosZ* genes in fertilized soil with diverse N sources. However, Domeignoz-Horta et al. [86] analyzed over 59,000 field measurements and concluded that the diversity of the *nosZ* is the most important factor explaining the variation in $N_2O$ emissions. Hallin et al. [90] demonstrated that a 50-year period of N fertilizer application significantly reduces potential denitrification rates and *nirK*, *nirS*, and *nosZ* gene abundance [70]. Similarly, Liang et al. [91] showed that

over 100 years of N fertilization reduces the temporal turnover of functional communities involved in denitrification. In contrast, several studies have found that N fertilization significantly increases *nir*K, *nir*S, and *nos*Z gene abundances [27,70].

The impact of N fertilization on soil greenhouse gas emissions has been well investigated and synthesized. Six meta-analyses have synthesized the impacts of N application on N processes and soil $N_2O$ emission. Carey et al. [69] examined 98 sets of measurement data from 33 articles to understand how N additions affect the abundances of *AOA* and *AOB* during nitrification and found that nitrification genes *AOA* and *AOB* responded differently to N fertilizer, with N applications having the most effect on *AOA* in croplands and on *AOB* in wildlands [35]. Ouyang et al. [27] investigated N functional genes (*nifH*, *AOA*, *AOB*, *nirS*, *nirK*, and *nosZ*) in response to N fertilization in agricultural ecosystems. They reported that, except for *nifH*, functional gene abundances increase during nitrification (*amoA*) and denitrification (*nirK*, *nirS*, and *nosZ*) when fertilized with N. In a meta-analysis by Li et al. [70], it was observed that prolonged N fertilization leads to a substantial 75.9% increase in potential denitrification activity. Additionally, there is an elevation in the proportions of *nirK*, *nirS*, and *nosZ* gene copies. Furthermore, the denitrification $N_2O/(N_2O + N_2)$ ratio and $nir(K + S)/nosZ$ ratio also experience an increase under prolonged N fertilization.

Our mega-analysis showed that N application tended to increase *AOA* (15%, 95% CI was [−0.8, 33.3]) and significantly enhanced *AOB* by 146.3%, *nirK* by 43.2%, and *nirS* by 31.8% (Table 2). It did not change MBC, *nifH*, and *nosZ* but increased MBN by 28.4%, soil $NH_4^+$-N by 33.9%, soil $NO_3^-$-N by 94.2%, and total N by 27.1% (Tables 1 and 2). All meta-analyses showed an increase in soil $N_2O$ emission, ranging from 121% to 258.8%, with a grand mean of increase in soil $N_2O$ emission by 153.2%. The strong enhancement of soil $N_2O$ emission was attributed to the changes in microbial community composition, stimulated N transformation processes, and elevated soil nutrient levels.

*5.2. Impacts of Biochar Applications*

Biochar, a recalcitrant carbonaceous biomass product generated through pyrolysis, has attracted considerable attention in agriculture for its potential to mitigate soil N losses and enhance N use efficiency [14,92,93]. Its application can enhance soil aeration, increase soil pH, promote microbial N immobilization, modify enzyme activities, and potentially affect nitrifier and denitrifier communities [94]. As reported by Zhang et al. [14] and Kaur et al. [25], the introduction of biochar into the soil has been found to elevate soil $NH_4^+$ and $NO_3^-$-N levels. This addition is associated with increased N mineralization, nitrification, $N_2$ fixation, and enhanced plant N uptake. Moreover, it also decreases N loss and $N_2O$ emissions. An increasing number of studies demonstrate that biochar amendments can alter soil microbial communities and N-cycling gene abundance [36,40,95,96]. For example, Ducey et al. [95] found that the incorporation of biochar leads to increased levels of *nifH* and *nirS*, and *amoA* and *nosZ* in soil [40,97]. However, some other studies have not reported significant effects of biochar addition on the abundance of N-cycling microbial genes, and a few have even reported a decrease in the abundance of *nifH* and N-cycling enzyme activity [98–100]. Recent research has highlighted how biochar amendment can modify soil pH, which, in turn, affects the abundance and diversity of N-cycling genes [40,90]. Van Zwieten et al. [101] suggested that biochar addition increases the abundance of *nosZ* transcripts, consequently reducing $N_2O$ emissions.

Quite some meta-analyses have been conducted and confirmed that biochar application can effectively reduce N losses, including N leaching and $N_2O$ emissions [14,102–104]. Kaur et al. [25] recently synthesized 18 meta-analyses and reported that biochar application reduces soil $N_2O$ emissions by 38%. But, only three meta-analyses synthesized genes involved in N processes [14,40,72]. Xiao et al. [40] reviewed 36 papers and found that biochar addition significantly increases the abundance of *AOA*, *nirK*, *nirS*, and *nosZ* by an average of 25.3%, 32.0%, 14.6%, and 17.0%, respectively, and reduced soil $NH_4^+$-N (15.5%), soil $NO_3^-$-N (−12.9%), and soil $N_2O$ emission (−14.6%). Zhang et al. [14] analyzed 131 field experiments and showed that biochar significantly enhances soil $NH_4^+$-N (5.3%) and $NO_3^-$-N

(3.7%) contents, N mineralization (15.3 or 13.5%), nitrification (48.5%), $N_2$ fixation (14.7%), and plant N uptake (18.3%) but reduced $N_2O$ emissions by 14.9%. Biochar application also increased the abundance of soil denitrifying/nitrifying genes (*amoA*, *narG*, *nirS/nirK + S*, and *nosZ*), the proportion of $N_2$ fixation bacteria, and N-acetyl-glucosaminidase activity by 18.6–87.6%. Meng et al. [72] also analyzed the impacts of biochar application on *AOA*, *AOB*, and soil $N_2O$ emission and found that biochar application enhanced *AOB* and reduced soil $N_2O$ emission.

Synthesizing these three meta-analyses, our mega-analysis showed that biochar application did not influence *nifH* and *AOB* but enhanced *AOA* by 23.0%, *nirS* by 19.1%, *nirK* by 28.3%, and *nosZ* by 17.1% (Table 2). In addition, biochar application enhanced $NH_4^+$-N by 6.0%, $NO_3^-$-N by 3.4%, and total N by 11.1% and reduced $NH_3$ emission by 34.0%, resulting in an 11.4% reduction in N leaching and 15.8% reduction in soil $N_2O$ emissions (Table 1). The reduction of soil $N_2O$ emissions with biochar application, despite the stimulation of N transformation genes and the enrichment of soil N levels, may be attributed to the complex interplay of various factors. Biochar's influence on microbial community composition, soil physical and chemical properties, and N retention may collectively contribute to a more balanced and controlled N cycling, reducing soil $N_2O$ emissions. The specific mechanisms likely depend on the unique conditions of the study site and the properties of the biochar used.

### 5.3. Impacts of Nitrification Inhibitor Usage

Nitrification inhibitors (NIs), such as dicyandiamide (DCD), 3,4-dimethylpyrazole phosphate (DMPP), and 2-chloro-6-(trichloromethyl) pyridine (nitrapyrin), have been commonly employed to mitigate $N_2O$ emissions by delaying the microbial oxidation of $NH_4^+$ to $NO_3^-$ in the soil and limiting nitrification and denitrification [30,105]. Previous individual studies have explored the influence of NIs on soil $N_2O$ emissions and associated functional gene and transcript abundances, and community structure. However, results have been inconsistent [30]. In general, *AOB* tends to dominate nitrification in neutral and alkaline soils, while *AOA* is more prevalent in acidic environments [106]. Increasing the $NH_4^+$ concentrations can enhance the nitrification activity of *AOB* [106], while *AOA* prefers environments with lower $NH_4^+$ concentrations. Furthermore, many studies showed that NIs effectively decreased the *AOB* population but not *AOA* [30,65,106]. In contrast, for alkaline paddy soil in China, nitrapyrin decreased the rates of nitrification and denitrification by limiting the abundances of *AOA* and *nirK*, respectively [30].

Recently, four meta-analyses have investigated the impact of NIs on the abundance of N-cycling genes and the release of $N_2O$ from the soil. Yin et al. [30] conducted a synthesis of 88 studies, revealing that the use of NIs significantly decreases the number of *AOB* genes, *nirS*, and *nirK* genes. NIs contributed to a 34.5% reduction in the activity of soil nitrifying enzymes (potential nitrification) and a 27.0% decrease in denitrifying enzymes (potential denitrification). Consequently, there is a notable decline of 63.6% in soil $N_2O$ emissions. Lei et al. [106] synthesized 48 papers and reported that NIs on average reduced 58.1% of $N_2O$ emissions and increased 71.4% of soil $NH_4^+$-N concentrations. The abundance of *AOB amoA* genes was dramatically reduced by about 50% with NI application in most soil types. Meng et al. [72] synthesized several studies related to NI on N pools and process genes and found that NI increased $NH_4^+$-N and total N but reduced potential nitrification and soil $N_2O$ emission. Guo et al. [5] synthesized 166 published papers, and N-cycling inhibitors decreased soil *AOB amoA* gene abundances (212%) and significantly decreased the *nirS* gene (39%). In general, NIs consistently exhibit a substantial reduction in the release of $N_2O$ from the soil, showing a negative correlation with the amounts of *nirK* and *nirS* genes. This consistent decrease in $N_2O$ emissions is a common finding across all meta-analyses.

Our mega-analysis showed that NIs did not change *AOA* but reduced *AOB* by 51.9% (ranged from −4.3 to −56.7% of four meta-analyses), as well as *nirK* (−20.0%) and *nirS* (−22.7%) (Table 2). NIs enhanced soil $NH_4^+$-N (87.8%) but reduced $NO_3^-$-N (−37.7%), leading to a reduction of soil $N_2O$ emission by 56.1% (from −51.7% to −62.7%) (Table 1).

### 5.4. Effects of Liming

Liming has the potential to reduce soil $N_2O$ emission through two primary mechanisms [73]. Firstly, it can achieve it by increasing the population of *nosZ*-type denitrifying bacteria while decreasing the ratio of fungi to bacteria. Both of these changes contribute to a lower $N_2O:N_2$ production ratio. Secondly, liming can also lower the amount of soil mineral N by promoting plant uptake. Zhang et al. [73] conducted a global meta-analysis using 1474 paired observations from 124 studies to explore the responses of GHG emissions to liming. They found that liming enhances nitrification by 62.7% and denitrification by 201.1%, increases soil $NO_3^-$-N by 55.8%, and reduces soil $N_2O$ emissions by 37.9%. Liming has been found to increase the abundance of *AOA* by 70.2%, *AOB* by 132.6%, *nik*S by 37.5%, and *nir*K by 142.0% (Table 2). Interestingly, liming does not change *nosZ* copy numbers. Given its significant influence on both $N_2O$ emissions and soil microbes, liming represents a potential strategy for mitigating soil $N_2O$ emissions.

### 5.5. Impacts of Microplastics

Microplastics pose a significant threat to ecosystem health, disrupting soil biological activities, and affecting biogeochemical cycles [74,107–109]. They can alter community structures of soil microorganisms and may ultimately impact the corresponding N process [9,72,74,110]. Functional microorganisms, particularly those with the *amoA* marker gene, play a crucial role in the denitrification process [74,111]. On the other hand, functional microorganisms with marker genes, including *narG*, *napA*, and *nirS*, are responsible for the denitrification process.

Several studies, including Gao et al. [112], Li et al. [110], and Zhang et al. [108], have explored the impact of microplastics on functional microorganisms involved in the N processes. The findings indicate varying effects on specific genes associated with nitrification and denitrification. With microplastics present, contrasting trends in the copy numbers of *nirS* have been observed, with some studies reporting an increase [113] and others a decrease [114]. Similarly, *amoA* gene sizes either remain constant or decrease. The divergent reactions of genes can complicate the understanding of how nitrification and denitrification impact $N_2O$ emissions.

Su et al. [74] conducted a meta-analysis using 60 published studies and found that in the presence of microplastics, $N_2O$ emissions surged by 140.4%, while nitrate reductase activities increased by 4.9%. The rate of denitrification rose by 17.8%, accompanied by a 10.6% increase in the number of genes responsible for denitrification (Tables 1 and 2). This suggests that microplastics may significantly enhance the genetic potential of microorganisms to carry out denitrification. Conversely, the nitrification rate and nitrifier genes exhibited minimal changes. The changes in N processes, especially the acceleration of denitrification, were identified as key contributors to increased $N_2O$ emissions. Microplastics may also create microenvironments that favor the growth and activity of denitrifying microorganisms, leading to remarkably increased $N_2O$ emissions.

### 5.6. Impacts of Crop Diversity

Diverse crops contribute significantly to both plant biomass and play a crucial role in shaping the functional microbial communities in the soil [115,116]. The introduction of crop diversity has been shown to increase the number of *nifH* gene copies in the soil and induce changes in microbial community structure [117,118]. In the context of intercropping legumes with non-legume plants, Chen et al. [119] found that it does not have a substantial impact on the amount of *nifH*. However, in mixed teak forests, the soil experiences a decrease in the amount of *AOB* alongside an increase in the number of *nosZ* genes. This phenomenon correlated with variations in total N and $NH_4^+$-N in the soil [120]. Moreover, agricultural systems with crop cycles, especially supplemented with inorganic N fertilizers, tend to increase the abundance of soil *AOB*, *nirK*, and *nosZ* genes, ultimately contributing to increased $N_2O$ emissions [121,122].

Using a meta-analysis, Hao et al. [31] found that soil *nifH*, *nirS*, *nirK*, and *narG* abundances were positively affected by the diversity of plant species, whereas the *amoA* and *nosZ* showed no response, based on 189 observations. In particular, crop diversity significantly reduced nitrification (−24.4%) and lowered soil $NH_4^+$-N (−11.4%) and $NO_3^-$-N (−6.8%), resulting in a reduction in soil $N_2O$ emissions (−19.9%) (Table 1). In addition, crop diversity enhanced *nifH* (33.8%), *narG* (17.8%), *nirS* (39.4%), and *nirK* (14.0%) (Table 2). Crop diversity enhanced the abundance of genes associated with N denitrification, reduced nitrification, lowered soil N concentrations, and significantly reduced soil $N_2O$ emissions. These findings highlight the potential of diverse plant species in agroecosystems to positively influence soil health and mitigate environmental impacts.

### 5.7. Impacts of Grazing

The response of N functional gene abundances to grazing has been inconsistent [8]. In an incubation experiment, Le Roux et al. [123] found that grazed soils exhibit higher levels of *AOA* and *AOB*, along with an elevated potential nitrification rate compared to control soils that have not been grazed [8]. In an alpine meadow in China, Zhang et al. [29] reported that grazing leads to increased $N_2O$ emissions and higher abundances of *AOA*, *AOB*, *nirK*, *nirS*, and *nosZ*. Conversely, Zhong et al. [124] reported that the moderate grazing of arid grassland does not affect the abundances of *narG* and *nosZ* or nitrification or denitrification rates. In the context of light grazing, Yin et al. [8] observed no significant effect on $N_2O$ emissions, the nitrification rate, or the denitrification rate compared to non-grazed land. However, under moderate to heavy grazing, there are notable changes in the abundance of key N functional genes. The amounts of *AOB* and *AOA* decrease, and there are sharp reductions in the amounts of *narG* and *nirS*. Interestingly, the abundance of *nosZ* remains unaffected. Using a laboratory incubation study, Pan et al. [122] demonstrated that heavy grazing decreases the abundance of *AOB* and the potential nitrification rate, while light grazing increases the abundance of *AOA*. In contrast, grazing over nine years decreases the abundance of *AOA*, *AOB*, *narG*, *nirS*, and *nirK* in semi-arid grassland [124]. Ding et al. [125] also reported that grazing reduces the abundance of *AOA*, *AOB*, and *nirK* in loamy-sand soil.

Yin et al. [8] conducted a meta-analysis with 83 published studies and found that heavy and moderate grazing reduced $N_2O$ emissions by 22–25%, nitrification rate by 23–37%, and denitrification rate by 44–48%, respectively, compared to the ungrazed condition. Moderate to heavy grazing intensities decreased the abundances of *AOB* by 40–47%. Heavy grazing also simultaneously decreased *AOA* by 34.9% (Table 2). Additionally, grazing significantly decreased the abundance of *narG* (−28%) and *nirS* (−35%) but did not affect the abundance of *nosZ* (Table 2). Livestock grazing at an appropriately moderate intensity is important for sustaining livestock production while contributing to greenhouse gas mitigation.

### 5.8. Impacts of Earthworms

While earthworms naturally inhibit the soil, incorporating sustainable agricultural practices such as increasing the return of organic matter to the soil, reducing tillage, adopting crop rotation, and avoiding the application of harmful chemicals can substantially enhance the presence of earthworms. Earthworms, often regarded as ecosystem engineers, play a crucial role in shaping soil health and ecosystem processes such as the N cycle [75]. The extent of their impact on the N cycle is closely related to their N-rich metabolic byproducts, the turnover of the N pool within the earthworm biomass, and the contribution of their decreased tissues [126,127]. Earthworms exert an indirect influence on the N cycle by changing the distribution of soil particles and incorporating pre-decomposed organic matter. Their activities, such as burrowing, contribute to an increase in the amount of macroaggregates, a factor that plays a crucial role in regulating N-cycling microorganisms in the soil [20]. Their burrowing activity further enhances N transformation through the input of organic materials into the root bioprocess [128]. Furthermore, earthworm intestinal tracts create a conducive environment for the survival of N-cycling microorganisms, stimu-

lating various N-cycle processes [129]. Earthworms have been found to exert significant effects on soil N-cycling microorganisms, such as the abundance of *amoA* gene of soil *AOB*, and significantly promote soil N-cycle processes, including denitrification, mineralization, and plant assimilation. In laboratory studies and experiments involving legume plants and in clay soils, earthworms are observed to significantly increase soil $N_2O$ emissions [75].

Xue et al. [75] conducted a meta-analysis with 130 publications and found that earthworms significantly enhanced soil $NH_4^+$-N (71.2%), $NO_3^-$-N (17.9%), MBN (13.7%), and soil $N_2O$ emission by 28.9% (Table 1). Earthworms also affected soil N-cycling microorganisms, including the *amoA* gene abundance of *AOB* (10.8%), and significantly promoted denitrification (38.6%) and mineralization (217.8%) (Tables 1 and 2). The presence of earthworms in the soil had complex effects on N dynamics and soil $N_2O$ emission. While earthworms actively enhanced soil N levels and MBN and promoted mineralization, they also led to an increase in soil $N_2O$ emissions. This meta-analysis reveals the positive impact of earthworms on the abundance of soil N and the available N content to soil microbes. These observed effects have the potential to alter the functions and services of ecosystems related to N cycling.

## 6. Regulating Factors on Soil $N_2O$ Emissions

Several factors have been identified as influencers of soil $N_2O$ emissions [9,29,40,58,72]. Notably, among these meta-analyses, three studies found that soil $N_2O$ was associated with climate factors such as mean annual temperature (MAT) and mean annual precipitation MAP (Table 3). In particular, soil $N_2O$ emission demonstrated a linear increase with MAT, while a concave relationship with soil MAP and soil moisture. Additionally, most of the studies found soil properties, especially soil available N ($NH_4^+$-N and $NO_3^-$-N) and C:N ratio, were associated with soil $N_2O$ emissions. Both positive and negative relationships between soil $N_2O$ emission and soil pH were found. Concerning N functional genes, soil $N_2O$ was predominantly found to have a negative correlation with *AOA* and a positive correlation with *AOB*. Furthermore, *nikS* and *nikS* were identified in some studies as significantly linked to soil $N_2O$ emissions (Table 3). However, studies also found that soil $N_2O$ emission is not closely related to nitrifier and denitrifier abundances [29]. It is important to note that additional studies are required to establish consensus results in these areas.

**Table 3.** Contributions of moderators for the responses of the soil $N_2O$ emission to different treatments. Values are regression slopes, correlation coefficients, or percentage of contributions.

| | Elevated $CO_2$ | | Nitrogen Fertilization | | Biochar Addition | | | Nitrification Inhibitor | | Liming | Grazing |
|---|---|---|---|---|---|---|---|---|---|---|---|
| | Du et al. [12] | You et al. [9] | Zhang et al. [14] | Meng et al. [72] | Zhang et al. [14] | Xiao et al. [40] | Guo et al. [5] | Lei et al. [106] | Yin et al. [30] | Zhang et al. [73] | Yin et al. [8] |
| **Climate** | | | | | | | | | | | |
| MAT | Linear decrease | 32.7 * | | | | | | | | | |
| MAP | Concave | 28.8 * | Nonlinear increase | | | | | | | | |
| **Soil** | | | | | | | | | | | |
| Soil moisture | Quadratic equation | | | | | | | | | | 5.35% * |
| Soil C:N ratio | | | Linear increase | | | | | | | | |
| $NH_4^+$-N | Linear decrease | | | | | | −0.33 | | | 0.214 | |
| $NO_3^-$-N | Quadratic increase | | | | | | 0.17 | | | | |
| Available N | | 21.0 * | | | | | | | | | 2.62% |
| SOM | | | | Linear decrease | | | | | | | |
| pH | | | Liner decrease | Linear increase | | | −0.63 | | | | |
| Soil texture | | | | | | | | | | | 0.63% |
| **Plant** | | | | | | | | | | | |
| Yield | | | | | | | | | | −0.99 * | |
| Vegetation type | | 8.6 | | | | | | | | | |

**Table 3.** *Cont.*

| | Elevated CO$_2$ | | Nitrogen Fertilization | | Biochar Addition | | | Nitrification Inhibitor | | Liming | Grazing |
|---|---|---|---|---|---|---|---|---|---|---|---|
| | Du et al. [12] | You et al. [9] | Zhang et al. [14] | Meng et al. [72] | Zhang et al. [14] | Xiao et al. [40] | Guo et al. [5] | Lei et al. [106] | Yin et al. [30] | Zhang et al. [73] | Yin et al. [8] |
| **N functional gene** | | | | | | | | | | | |
| AOA | | 11.6 * | −1.623 * | | −0.070 | −0.28 * | −0.37 | −0.06 | 0.29 | −0.90 * | |
| AOB | | 10.7 * | −0.653 * | | 0.849 * | −0.001 | 0.07 | 0.42 ** | 0.31 * | 0.99 * | |
| nosZ | | 30.7 * | | | 0.363 | −0.12 | | | 0.29 | −1.43 * | |
| nirK | | | | | 0.965 * | 0.20 | | | 0.50 * | | |
| nirS | | | | | 0.885 * | 0.41 * | | | 0.13 | | |
| narG | | | | | 0.808 * | | | | 0.69 | | |
| Nitrification | | | | Nonlinear * | | | | | | | |
| Denitrification | | | | | | | 0.24 | | | −0.25 * | |
| Yield-scaled NH$_3$ | | | | | | | 0.48 | | | | |
| **Treatment** | | | | | | | | | | | |
| N application rate | | 12.1 * | | Linear increase | | | | | | | |
| Nitrogen form | | 8.2 | | | | | | | | | |
| Grazing intensity | | | | | | | | | | | 13.26% * |
| Grazing duration | | | | | | | | | | | 0.58% |

Note: * indicates $p < 0.05$ and ** $p < 0.01$. MAP: mean annual precipitation; MAT: mean annual temperature. *SOM: Soil organic matter. AOA: amoA* genes from ammonia-oxidizing archaea. *AOB*: *amoA* genes from ammonia-oxidizing bacteria. *narG:* nitrate reductase. *nirS* and *nirK:* nitrite reductase. *nosZ:* N$_2$O reductase.

### 7. Conclusions and Future Research Directions

$N_2O$, a potent greenhouse gas originating from soil microbial processes like nitrification and denitrification, is significantly influenced by diverse factors that are shaped by climate change and agricultural practices. In this mega-analysis, we quantified their impacts on soil $N_2O$ emissions based on 25 meta-analyses. The findings revealed that global warming substantially increased soil $N_2O$ emissions by 159.7%, attributed to enhanced rates of both soil nitrification and denitrification. Elevated $CO_2$ stimulated soil $N_2O$ emissions by 40.6%, potentially linked to changes in denitrification. The application of N fertilization emerged as a significant contributor, increasing soil $N_2O$ emissions by 153.2%, largely associated with elevated abundances of *AOB*, *nirS*, and *nirK*, along with higher soil N levels. On the other hand, biochar addition reduced soil $N_2O$ emission by 15.8%. Interestingly, microplastics had adverse impacts, stimulating soil $N_2O$ emissions by 140.4%, primarily due to an intensified denitrification process. Additionally, earthworm activity was found to enhance soil $N_2O$ emissions by 28.9% as earthworms actively enhanced soil N levels and MBN and promoted mineralization.

Despite the valuable insights provided by numerous previous studies for mitigating soil $N_2O$ emissions and formulating effective management practices, critical knowledge gaps persist. The spatial and temporal variations in soil $N_2O$ emissions, ranging from small plots to ecosystems, on regional and global scales, remain unclear. Establishing a comprehensive network for monitoring soil $N_2O$ emissions across diverse ecosystems at large spatial scales over extended periods is crucial. This initiative not only aids in accurate emission estimates but also contributes valuable data for regional, national, and global ecosystem modeling.

Microbial process studies play a pivotal role in understanding the mechanisms of soil $N_2O$ emissions and developing reduction strategies. Focused investigations into the microbial processes involved in $N_2O$ production can pinpoint key microorganisms responsible for $N_2O$ emissions. Utilizing molecular techniques like DNA sequencing and metagenomics reveals microbial community composition and diversity, shedding light on changes in soil $N_2O$ emissions. Additionally, identifying $N_2O$-reducing microorganisms capable of converting $N_2O$ to $N_2$ holds promise for emission mitigation. Other potential strategies include optimizing fertilizer application rates, employing nitrification inhibitors, promoting specific microbial communities, implementing sustainable practices such as biochar application, and fostering a harmonious balance between agricultural productivity and environmental sustainability in the face of evolving climate challenges.

**Supplementary Materials:** The following supporting information can be downloaded at: https://www.mdpi.com/article/10.3390/agriculture14020240/s1, Figure S1: Number of publications based on the topic search of "meta-analysis" and "$N_2O$ emission" using Web of Science.

**Author Contributions:** Conceptualization, D.H.; methodology, D.H., A.R., L.K. and J.C.; writing—original draft preparation, D.H.; writing—review and editing, D.H., A.R., L.K. and J.C.; funding acquisition, D.H. All authors have read and agreed to the published version of the manuscript.

**Funding:** This research was funded by the U.S. Department of Agriculture (USDA) CBG project TENX12899 (2022-38821-37341) and the National Science Foundation (NSF) EiR project (2000058).

**Data Availability Statement:** No new data were created. Data are contained within this article.

**Conflicts of Interest:** The authors declare that there are no conflicts of interest regarding the publication of this paper.

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
