# Peer review of "Impacts of Climate Change and Agricultural Practices on Nitrogen Processes, Genes, and Soil Nitrous Oxide Emissions: A Quantitative Review of Meta-Analyses"

_agriculture, doi:10.3390/agriculture14020240_

Round 1

Reviewer 1 Report

Comments and Suggestions for Authors

This review paper evaluated soil N-cycling processes, microbial genes, and N2O emission following climate change and agricultural practices, based on meta-analysis (meta-meta-analysis). The results that the authors summarized are systematic and comprehensive, which advance our understanding of soil N cycling processes and the underlying mechanisms. I have several comments below, which I hope that will help the authors improve the manuscript before publication.

In the Abstract section, it is better to show the impacts of other global change factors and agricultural practices, although some of them may have no significant effects.

Climate change usually refers to warming, changes in precipitation pattern, etc.. While atmospheric N deposition is closely related to anthropogenic activities, and it may not belong to climate change.

Line 67-72. How many relevant meta-analyses have been published by far? It is useful if you can add a relevant figure (e.g., numbers, years, research topic, etc.) through Web of Science.

Line 72. How do you consider the selection of the published meta-analyses? Many of those meta-analyses with similar research topics may have a part of the same dataset or data source. Will these affect the results of calculation by mega-analysis? How to reduce the impacts from such potential weightiness imbalance?

Table 1. Nitrogen fixation have many sources, including symbiotic and non-symbiotic pathways. Is the dataset limited to soil N fixation or legume N fixation?

Line 204. The impacts of global warming on soil N-cycling processes can vary with ecosystem types and climatic zones. For example, those in forest soils can have different responses with those in grassland soils. Cold regions may show more sensitive responses to warming than warm regions. While I find that such information is less discussed.

Line 179. It is not clear what the ecosystem types of the dataset were collected? Was it limited to agricultural systems only? Some climate change factors (e.g., warming, N deposition) may be applied to natural ecosystems, and some (e.g., grazing) can be from grasslands. The ecosystem types of the dataset should be more clearly defined.

Line 333. I am not sure how to differentiate the difference of N deposition and N fertilization. As shown in Fig. 1, N deposition is a global change factor while N fertilization is a type of agricultural practice. Whether the impacts and mechanisms are similar for these two pathways?

Fig. 1. It seems that microplastics and earthworms may not belong to ‘agricultural practices’.

Line 540 This section is not directly linked to agricultural practices, but a natural process. I suggest it can be removed or combined into other relevant section.

Line 566 The section of regulating factors is a bit poor. As we known, there have been a lot of factors (climatic conditions, nutrient status, microbial community, etc.) regulating soil N2O emission, but the current section is less discussed.

Author Response

Reviewer: 1

Comments to the Author
This review paper evaluated soil N-cycling processes, microbial genes, and N2O emission following climate change and agricultural practices, based on meta-analysis (meta-meta-analysis). The results that the authors summarized are systematic and comprehensive, which advance our understanding of soil N cycling processes and the underlying mechanisms. I have several comments below, which I hope that will help the authors improve the manuscript before publication.

Response: We thank this reviewer for the positive comments and constructive suggestions on our manuscript. All the suggestions by the reviewer have been incorporated into this revision. We appreciate the reviewers’ comments that significantly improved our manuscript.

In the Abstract section, it is better to show the impacts of other global change factors and agricultural practices, although some of them may have no significant effects.

Response: We initially included more results from other global change factors and agricultural practices. But due to the limitation of the words (200 words) in the abstract, we revised the abstract by removing some results and only keeping the key results. Right now, there are 199 words in the abstract.

Climate change usually refers to warming, changes in precipitation pattern, etc.. While atmospheric N deposition is closely related to anthropogenic activities, and it may not belong to climate change.

Response: We agree with the reviewer and removed “Nitrogen deposition” from the Climate change in Fig. 1.

Line 67-72. How many relevant meta-analyses have been published by far? It is useful if you can add a relevant figure (e.g., numbers, years, research topic, etc.) through Web of Science.

Response: The number of meta-analyses was provided in Line 184-187. In this revision, we re-checked the publications using the Web of Science, and added a supplemental figure showing numbers of publications over the years based on Web of Science (Lines 183 in the manuscript with highlighted changes, Figure S1).

Line 72. How do you consider the selection of the published meta-analyses? Many of those meta-analyses with similar research topics may have a part of the same dataset or data source. Will these affect the results of calculation by mega-analysis? How to reduce the impacts from such potential weightiness imbalance?

Response: The reviewer raised a very good question related to data selection and weighting factor. In this study, we calculated grand mean response ratio using sample size as a weight factor, following our previous study (Kaur et al. 2023 [25]). In Kaur et al. (2023), we discussed the issues related to the impacts of the duplicated datapoints and considered our estimation as a conservative one. To verify the impacts of data duplication, we have collected data from individual studies used in the 18 meta-analyses and are working on a comprehensive meta-analysis now.

Table 1. Nitrogen fixation have many sources, including symbiotic and non-symbiotic pathways. Is the dataset limited to soil N fixation or legume N fixation?

Response: We used the results from meta-analyses and did not separate soil N fixation or legume N fixation in this study.

Line 204. The impacts of global warming on soil N-cycling processes can vary with ecosystem types and climatic zones. For example, those in forest soils can have different responses with those in grassland soils. Cold regions may show more sensitive responses to warming than warm regions. While I find that such information is less discussed.

Response: We agree with the reviewer and added the above statements in the revised manuscript as suggested (Lines 213-215).

Line 179. It is not clear what the ecosystem types of the dataset were collected? Was it limited to agricultural systems only? Some climate change factors (e.g., warming, N deposition) may be applied to natural ecosystems, and some (e.g., grazing) can be from grasslands. The ecosystem types of the dataset should be more clearly defined.

Response: We did not limit the ecosystem type in this study. But for the agricultural practices, most of these were applied in agroecosystems. For climate change, we observed more studies in natural ecosystems, particularly grasslands. Since we are interested in their impacts on N process, specifically genes related to nitrification and denitrification, we found only 25 meta-analyses and this mega-analysis is based on these 25 meta-analyses.

Line 333. I am not sure how to differentiate the difference of N deposition and N fertilization. As shown in Fig. 1, N deposition is a global change factor while N fertilization is a type of agricultural practice. Whether the impacts and mechanisms are similar for these two pathways?

Response: This is a good question. Nitrogen deposition is a global change issue, but not part of climate change. In this revision, we removed it from the Climate change. It is difficult to distinguish the impacts of N fertilization and N deposition, as N deposition is just N fertilization at a relatively low application rate. In this study, we considered N deposition as part of N fertilization.

Fig. 1. It seems that microplastics and earthworms may not belong to ‘agricultural practices’.

Response: We understand the concern raised by the reviewer. We think that microplastics is the results of the use of plastics in agriculture practices (mostly plastic mulch). As a result, we put Microplastics into Agricultural practices. Regarding earthworms, they play a crucial role in agricultural practices. They help improve soil structure by burrowing through the soil and creating channels for air and water to reach plant roots, and consume organic matter. They function as agricultural practices. Thus, we did not separate it from others in this study.

Line 540 This section is not directly linked to agricultural practices, but a natural process. I suggest it can be removed or combined into other relevant section.

Response: We agree with the review that earthworms naturally exist in soil, but farmers can actively encourage and support their presence as part of sustainable agricultural practices, such as adding more organic matter to soil, reducing tillage, implementing crop rotation, and avoiding harmful chemical application to soil. We added these to the revised manuscript (Lines 553-556). We hope that this part is more closely linked to agricultural practice now.

Line 566 The section of regulating factors is a bit poor. As we known, there have been a lot of factors (climatic conditions, nutrient status, microbial community, etc.) regulating soil N2O emission, but the current section is less discussed.

Response: We understand the reviewer’s concern here and totally agree with the reviewer that many factors could regulate soil N2O emission. Here, we summarized the results of these meta-analyses, and showed that indeed, different factors, from climatic factors to soil, microbial and functional genes, influenced soil N2O emissions. But no conclusive results can be generated from different treatments. We call for more studies so we can understand better how microbial nitrification and denitrification could contribute to soil N2O emissions.

Reviewer 2 Report

Comments and Suggestions for Authors

The authors adopted mega-analysis approach to quantify the impact of climate change and agricultural activities on soil nitrogen cycle genes, processes, and N2O emission. In this comprehensive review, they observed a significant increase in N2O emission with warming and elevated CO2, whereas the addition of biochar resulted in a substantial reduction in N2O emission. Overall, the manuscript is well-written, and its organizational structure is logical. Some minor issues, however, need to be fixed for clarity.

Specific comments:

1.     In the Abstract section, some descriptions are vague. For instance, there is a lack of quantitative results concerning the impact of microplastics on the nitrogen cycle process (Lines 20-21).

2.     Some statements in the Introduction section lack adequate references. For example, in Lines 70-71, the citation of previous meta-analysis results should be appropriately referenced.

3.     There is uncertainty regarding whether the gene napA belongs to the denitrification process. It is generally considered a part of the assimilatory N reduction pathway (Lines 123-125).

4.     In the Result section, the description is inconsistent with the corresponding tables. For instance, in Lines 242-243, the results should be corresponded to Table 2, but the authors’ citation refers to Table 1. Similarly, in Lines 243-245, the results are from Table 1, but the citation is Table 2.

5.     Some results may require further discussion to enhance persuasiveness. For instance, in Lines 280-284, concerning the impact of elevated CO2 on the nitrification process, the results show that elevated CO2 has no significant effect on genes associated with nitrification, yet it reduces the corresponding enzymes activity, increasing the nitrification rate. This description may require further discussion for clarity.

6.     In Line 308, gene ‘amoA’ should be italicized. It is recommended to perform a thorough check of gene formatting across the entire manuscript.

7.     Regarding the effect of precipitation changes on the nitrogen cycle process, discrepancies with previous meta-analyses need an appropriate explanation.

8.     Tables are preferably formatted as triple-line tables.

9.     It is better to make the Conclusion concise.

Comments on the Quality of English Language

Minor editing of English language is required.

Author Response

Reviewer: 2

Comments to the Author
The authors adopted mega-analysis approach to quantify the impact of climate change and agricultural activities on soil nitrogen cycle genes, processes, and N2O emission. In this comprehensive review, they observed a significant increase in N2O emission with warming and elevated CO2, whereas the addition of biochar resulted in a substantial reduction in N2O emission. Overall, the manuscript is well-written, and its organizational structure is logical. Some minor issues, however, need to be fixed for clarity.

Response: We thank this reviewer for the very positive comments. In this revision, we have carefully studied the reviewer’s comments and suggestions, and incorporated them into the revised manuscript.

Specific comments:

  1. In the Abstract section, some descriptions are vague. For instance, there is a lack of quantitative results concerning the impact of microplastics on the nitrogen cycle process (Lines 20-21).

Response: We added quantitative changes of soil N2O emission in the abstract (Line 21, in the manuscript with highlighted changes).

  1. Some statements in the Introduction section lack adequate references. For example, in Lines 70-71, the citation of previous meta-analysis results should be appropriately referenced.

Response: We added several references for these statements (Lines 70, 71).

  1. There is uncertainty regarding whether the gene napA belongs to the denitrification process. It is generally considered a part of the assimilatory N reduction pathway (Lines 123-125).

Response: We thank this reviewer for pointing this out. In the revised manuscript, we added this statement (Lines 127-128)

  1. In the Result section, the description is inconsistent with the corresponding tables. For instance, in Lines 242-243, the results should be corresponded to Table 2, but the authors’ citation refers to Table 1. Similarly, in Lines 243-245, the results are from Table 1, but the citation is Table 2.

Response: Sorry for the confusion here. In this revision, we double-checked all table citations and they are correctly cited now.

  1. Some results may require further discussion to enhance persuasiveness. For instance, in Lines 280-284, concerning the impact of elevated COon the nitrification process, the results show that elevated CO2has no significant effect on genes associated with nitrification, yet it reduces the corresponding enzymes activity, increasing the nitrification rate. This description may require further discussion for clarity.

Response: In this revision, we added more explanation to our results related to the impacts of climate changes and agricultural practices, including warming (Lines 252-256), CO2 (Lines 295-301), precipitation (Lines 343-348), N application (Lines 404-406), Biochar (Lines 447-453), microplastics (Lines 528-531), crop diversity (Lines 550-553), and earthworms (Lines 607-610). Regarding the results of CO2 impacts, we explained the inconsistent results: “While there was an increase in soil N2O emissions, possibly associated with changes in denitrification (as indicated by increased nirS abundance), there were mixed effects on nitrification-related parameters. The increased MBN, net nitrification rate, and NO3--N levels indicate a stimulation of N cycling processes, but the reduction in nitrifying enzymes suggests a potential deceleration of ammonia oxidation. These findings highlight the need to consider multiple factors influencing soil N dynamics in the context of elevated CO2.” (Lines 295-301).

  1. In Line 308, gene ‘amoA’ should be italicized. It is recommended to perform a thorough check of gene formatting across the entire manuscript.

Response: We checked the formatting for the whole manuscript, and all genes are italicized now.

  1. Regarding the effect of precipitation changes on the nitrogen cycle process, discrepancies with previous meta-analyses need an appropriate explanation.

Response: We added more explanations in this revision. The discrepancies could be caused by different ecosystems included in data analysis. For example, Yan et al. reported that precipitation increases enhanced soil N2O emissions by 128.3% in temperate forests and by 179.6% in boreal forests, but did not influence soil N2O emission in grasslands. The impacts of decreased precipitation also varied in different ecosystems, ranging from no effect in subtropical forest, to -24.3% in temperate forests and -92.6% in grasslands (Lines 331-335).

  1. Tables are preferably formatted as triple-line tables.

Response: We understand the table format requirement. We did not remove some horizontal lines as we feel that these lines help readers align the values, particularly due to not all values exist. We will be glad to remove these lines if the journal office does not allow them.

  1. It is better to make the Conclusion concise.

Response: We revised the conclusion and deleted two sentences in the conclusion.

Reviewer 3 Report

Comments and Suggestions for Authors

A good study, synthesising a good number of studies to explore N2O emissions and underlying key drivers, which is an appropriate topic for the journal.

What is missing largely is the explanation to help the reader to improve their mechanistic understanding on how each factors contribute to increase/decrease/remain unchanged processes. It is important to know, when you say microplstics increased N2O emissions by 140% ( for example) how mechanistically (physically, chemically or biologically) microplastics can improve N2O emissions that much. Such detailed explanation will greatly help from this type of meta or mega analysis where readers see mostly numbers but get limited insight on how they occur.

The conclusions seem to be very descriptive with no reference to your major findings. Your most key findings should be enumerated here for a reader, who sometimes read the conclusions only in your paper to grab the biggest fish in your net.

Comments on the Quality of English Language

I have attached some editorial comments on the paper. I basically see a sloppy writing at the first part of the paper, but a good language in the latter part of the paper.

Therefore, I recommend to improve the abstract and the introduction of the paper as there are many improper/awkward words frequently used in the text.

Author Response

Reviewer: 3

Comments to the Author
A good study, synthesising a good number of studies to explore N2O emissions and underlying key drivers, which is an appropriate topic for the journal.
Response: We thank this reviewer for the very positive comments and constructive suggestions which have been incorporated into the revised manuscript.

What is missing largely is the explanation to help the reader to improve their mechanistic understanding on how each factors contribute to increase/decrease/remain unchanged processes. It is important to know, when you say microplstics increased N2O emissions by 140% ( for example) how mechanistically (physically, chemically or biologically) microplastics can improve N2O emissions that much. Such detailed explanation will greatly help from this type of meta or mega analysis where readers see mostly numbers but get limited insight on how they occur.
Response: In this revision, we added more explanation to the results of major climatic factors and agricultural practices [warming (Lines 252-256, in the manuscript with highlighted changes), CO2 (Lines 295-301), precipitation (Lines 343-348), N application (Lines 404-406), Biochar (Lines 447-453), microplastics (Lines 523-524; 528-531), crop diversity (Lines 550-553), and earthworms (Lines 607-610)]. Regarding the impact of microplastics, we added our explanation: “This suggests that microplastics may significantly enhance the genetic potential of microorganisms to carry out denitrification. Conversely, the nitrification rate and nitrifier genes exhibited minimal changes. The changes in N processes, especially the acceleration of denitrification, were identified as key contributors to increased N2O emissions. Microplastics may also create microenvironments that favor the growth and activity of denitrifying microorganisms, leading to remarkedly increased N2O emissions.” (Lines 523-531).

The conclusions seem to be very descriptive with no reference to your major findings. Your most key findings should be enumerated here for a reader, who sometimes read the conclusions only in your paper to grab the biggest fish in your net.

Response: We thank this review for the suggestion. In this revision, we replaced the general description of the results with major quantitative findings (Lines 641-652).